# Exploring the Magnetic and Electrocatalytic Properties of Amorphous MnB Nanoflakes

**DOI:** 10.3390/nano13020300

**Published:** 2023-01-11

**Authors:** Boxiao Fu, Vasileios Tzitzios, Qiancheng Zhang, Brian Rodriguez, Michael Pissas, Maria Veronica Sofianos

**Affiliations:** 1School of Chemical and Bioprocess Engineering, University College Dublin, Belfield, Dublin 4, Ireland; 2Institute of Nanoscience and Nanotechnology, National Centre for Scientific Research “Demokritos”, 15310 Athens, Greece; 3School of Physics, Conway Institute of Biomolecular and Biomedical Research, University College Dublin, Belfield, Dublin 4, Ireland

**Keywords:** manganese borides, 2D nanoparticles, magnetism, electrocatalysis, hydrogen evolution reaction (HER), oxygen evolution reaction (OER)

## Abstract

Two-dimensional (2D) metal borides are a class of ceramic materials with diverse structural and topological properties. These diverse material properties of metal borides are what forms the basis of their interdisciplinarity and their applicability in various research fields. In this study, we highlight which fundamental and practical parameters need to be taken into consideration when designing nanomaterials for specific applications. A simple one-pot chemical reduction method was applied for the synthesis of manganese mono-boride nanoflakes at room temperature. How the specific surface area and boron-content of the as-synthesized manganese mono-boride nanoflakes influence their magnetic and electrocatalytic properties is reported. The sample with the highest specific surface area and boron content demonstrated the best magnetic and electrocatalytic properties in the HER. Whereas the sample with the lowest specific surface area and boron content exhibited the best electric conductivity and electrocatalytic properties in the OER.

## 1. Introduction

Two-dimensional (2D) materials have attracted great attention across the research community due to their interdisciplinarity and their applicability in various fields, such as chemistry, physics, engineering, and nanotechnology [1]. Among the most attractive class of 2D ceramic materials are transition metal borides as they possess superior properties and, therefore, are utilized in a broad range of applications. They have high melting points and hardness, they are thermodynamically stable, they have excellent electric conductivity and superconductivity, and they are catalytically active and corrosive resistant [1,2,3,4]. Hence, they are extensively used in sensing, electronics, catalysis, energy harvesting, ion transport, superconductivity, hydrogen generation, and storage [5,6,7,8,9]. Their stoichiometry varies from compounds with low a boron content similar to metal mono-borides (MB) to compounds with a very high boron content (MB_99_). The spatial distribution of boron atoms, in transition metal borides, can differ significantly. Boron atoms can be either isolated or paired, they can form chains or even three-dimensional connected networks as observed in transition metal borides with a high boron concentration [10]. This spatial distribution of boron atoms may be responsible for their superior properties, and their broad range of applications.

A plethora of synthesis methods have been reported in the literature for the production of either bulk or nano transition metal borides [1,10]. For the synthesis of bulk transition metal borides, high-temperatures are required ranging from 1200 to 2000 °C, whereas nano transition metal borides can be synthesized at room temperature and are predominantly amorphous with a flaky structure [11,12,13]. Among the reported high-temperature synthesis methods, the reactive process is considered the simplest. In this process, powders of elemental boron and the desired transition metal are mixed together and are then heated under an inert atmosphere, in this was the formation of oxide impurities are avoided during the reaction process [14,15]. For the synthesis of nano transition metal borides at room temperature, the most commonly used method is the chemical reduction in aqueous transition metal chlorides by sodium borohydride (NaBH_4_) in the presence of sodium hydroxide (NaOH) [16].

One of the most promising transition metal borides is manganese boride (Mn_x_B_y_) as it exhibits both a high electric and thermal conductivity, a high melting point, low compressibility, high hardness, and wear resistance and room temperature magnetic properties [17,18,19,20]. Its material properties are directly linked with its boron content. For examples, a low-boron content contributes to general soft mechanical properties, whereas a high-boron content contributes to low ferromagnetic properties [21]. Thus, having the correct boron content in manganese boride is highly important in order to achieve the enhanced material properties for the desired applications.

This study explores how the specific surface area and the boron content of 2D manganese mono-boride nanoparticles influence their magnetic and electrocatalytic properties. A simple one-pot chemical reduction method was applied to synthesize the Mn_1.03_B_0.97_ and Mn_1.21_B_0.79_ nanoflakes, using NaBH_4_, NaOH, and MnCl_2_ as the precursors. The manganese boride nanoflakes with the highest specific surface area and boron content exhibited the best magnetic properties and electrocatalytic performance in the hydrogen evolution reaction (HER). In contrast, the manganese boride nanoflakes with the lowest specific surface area and boron content showed the best electric conductivity and electrocatalytic performance in the oxygen evolution reaction (OER); demonstrating that the material properties of MnB, which influence the performance of the HER and OER are the exact opposite. Hence, this study emphasizes the significance of the fundamental and practical parameters that need to be considered when designing nanomaterials for specific applications.

## 2. Materials and Methods

### 2.1. Synthesis of MnB Nanoflakes

A simple one-pot chemical reduction method (see Figure 1) was used to synthesize the MnB nanoflakes. In detail, two aqueous solutions containing 0.27 M each of manganese chloride (MnCl_2_, ≥99% Sigma-Aldrich, Wicklow, Ireland) were made while stirring vigorously at room temperature. Next, sodium borohydride (NaBH_4_, ≥98% Sigma-Aldrich, Wicklow, Ireland) and sodium hydroxide (NaOH, ≥97% Sigma-Aldrich, Wicklow, Ireland) in a 10:1 molar ratio were mixed with distilled water at two concentrations (0.5 and 1 M NaBH_4_) and were carefully titrated to the above MnCl_2_ aqueous solutions. As soon as the generated bubbles stopped, the black precipitates from each synthesis were collected using a centrifuge. The two black powder products were rinsed two times with distilled water and once with ethanol, making sure that any impurities and/or unreacted compounds were removed, and then dried under vacuum at 70 °C (Table 1).

### 2.2. Characterization Techniques

Powder X-ray diffraction (XRD) was applied for phase analysis of the as-prepared MnB nanoflakes using a Siemens D500 (40 kV, 30 mA) diffractometer from Germany with Cu-Ka radiation (8.04 keV). The measured 2*θ* range (10°–80°) was scanned with a 0.03 step size, at 2 s/step and a rotational speed of 30 rpm. The XRD characterization could not be performed on the nanoflakes after electrocatalysis, since the mass used for all electrochemical measurements was 0.025 mg which is below the instrument’s resolution.

Morphological observations of the as-prepared MnB nanoflakes were carried out with a FEI Tecnai 20 transmission electron microscope (TEM). The nanoflakes, prior to imaging, were mixed with ethanol, ultrasonicated for 20 min, and then drop casted directly onto a carbon coated Cu TEM grid.

Their specific surface area was calculated by Small-Angle X-ray Scattering (SAXS) using a Bruker Nanostar instrument equipped with an Excillium MetalJet source (GaKα, λ = 1.3402 Å). The MnB nanoflake powder samples were compacted in a transmission geometry using tape. All the measurements were conducted in vacuum, and the background was removed and converted on an absolute scale using a NIST SRM3600 glassy carbon standard [22]. The two scattering patterns were modelled using the unified model [23] in Irena [24]. The specific surface area (SSA) was calculated from the high-q Porod region (level 4 of power slope) from the equation below:(1)$$SSA=\frac{B}{2\pi \delta \Delta \rho^{2}}$$
where *B* is the Porod constant as refined in the unified fit, *δ* is the crystallographic density, and Δ*ρ^2^* is the scattering contrast.

The chemical analysis of the as-prepared MnB nanoflakes was performed with a Kratos AXIS Ultra DLD X-ray photoelectron spectrometer (XPS) in ultra-high vacuum using an Al-Kα X-ray source (1486.7 eV). The data were processed using the Casa XPS software and calibrated using the surface adventitious C 1s peak at 284.5 eV. The XPS characterization of the nanoflakes after electrocatalysis could not be undertaken since the mass of the catalyst used was too small, being under the instrument’s detection limit, as previously mentioned.

The dc- and ac-magnetic measurements were carried out on a Physical Properties Measuring system (PPMS, Quantum Design, Ireland) using the ACMS measuring option. The ac-magnetic susceptibility data, as a function of temperature, were collected in a zero and 1 kOe external magnetic field. The amplitude and the frequency of the external ac-magnetic field were 10 Oe and 1111 Hz, respectively. The dc-magnetic moment measurements as a function of temperature were collected in zero field cooled (ZFC) and field cooled (FC) modes.

Prior to performing both Atomic Force Microscopy (AFM) and Conductive-Atomic Force Microscopy (C-AFM) and obtaining all the electrochemical measurements, an ink containing the MnB nanoflakes was prepared by dispersing 5 mg of the MnB powder in 5% Nafion (20 μL, Sigma-Aldrich, Ireland), with 490 μL of ethanol and 490 μL of ultrapure water then ultrasonicated for 20 min.

AFM and C-AFM were performed on the MnB nanoflakes. Prior to their topographical and electrical characterization, a droplet of the MnB ink was placed on a conductive indium tin oxide (ITO) coated glass cover slip and was allowed to dry at room temperature. Height images were obtained using an Atomic Force Microscope (AFM; Asylum Research, Oxford, UK, MFP-3D) equipped with a Si probe (Nanosensors, Switzerland, PPP-NCH) with a nominal resonance frequency of 330 kHz, a tip radius less than 7 nm, and a spring constant of 42 N/m. The conductive atomic force microscopy (C-AFM) investigations were carried out using the same AFM equipped with a C-AFM cantilever holder (Asylum Research, ORCA with 2 nA/V sensitivity) and a solid Pt AFM probe (Rocky Mountain Nanotechnology, USA, RMN-25PT300B) with a nominal resonance frequency, tip radius, and spring constant of 20 kHz, 20 nm and 18 N/m, respectively. The current-voltage (IV) spectroscopic measurements were conducted using the contact mode with a voltage range of 0.6 to −0.6 V and a 1 Hz 4 cycle triangle waveform, which was initially ramped up from 0 V and applied from the ORCA holder to the sample through the ITO electrode with the current being measured in the ORCA holder. Before the IV measurements, the current was zeroed by applying a small offset. The voltage range was selected to reduce the chance of current saturation. The current was saturated at 20 nA when the tip contacted the ITO in this voltage range. The reported sample resistances were determined as the inverse slope of the current, under the assumption that other resistances were zero. Measurements were made in different locations across each sample, and 20 IV sweeps were collected.

A three-electrode cell connected to an Autolab potentiostat (PGSTAT204 with a FRA32M Module, Metrohm, Carlo, Ireland) interfaced to a PC with Nova 2.0 software was used for performing the electrochemical measurements. A rotating disk glassy carbon (GC) electrode (3 mm diameter, Metrohm) was used as the working electrode, a silver/silver chloride (AgCl, 3 M KCl, Metrohm) electrode was used as the reference electrode, and a graphite rod was used as the counter electrode (MW-4131, BASi, Wimbledon, UK). Plain distilled water (pH 7) was used as the electrolyte. Prior to all electrochemical measurements, 5 μL of the ink containing 0.025 mg of MnB nanoflakes, was drop-casted onto the polished glassy carbon electrode with a 0.07 cm^2^ surface area. The film formed contained 0.36 mg/cm^2^ of MnB nanoflakes. The glassy carbon electrode was polished before drop-casting using a polishing cloth and aluminum oxide powder (grain size 0.3 µm). During the electrochemical measurements, the working electrode was rotated at 2000 rpm by a rotation unit (Metrohm) to eliminate bubbles. All measured potentials were converted to the reversible hydrogen electrode (RHE), according to the equation $E_{\mathrm{RHE}}=E_{\mathrm{Ag}/\mathrm{AgCl}}+0.207+0.059\;pH$. A borate buffer solution (pH 7) was used by mixing 50 mL of (0.2M KH_2_PO_4_, >99% Sigma-Aldrich) and 29.63 mL of sodium hydroxide (0.2M NaOH, ≥97% Sigma-Aldrich) in 120 mL of distilled water.

The electrochemical impedance spectroscopy (EIS) was performed at the open circuit potential (OCP) in the frequency 100 kHz–1 Hz. The resistance of the electrolyte Rs was determined from the Nyquist plot (Appendix A) and applied for ohmic drop correction following the equation $E_{c}=E_{e}-iR_{s}$, where *E_c_* is the corrected potential and *E_e_* is the experimental potential (Appendix A) [25]. The raw data with no $iR_{s}$ correction is not presented in this work. The HER and OER activity were evaluated using a linear sweep voltammetry (LSV) at a scan rate of 1 mVs^−1^ under a N_2_ gas (>99.999%, BOC) atmosphere [22]. The stability performance of the MnB nanoflakes was studied by performing 100 scans (cycles) at a scan rate of 1 mVs^−1^.

## 3. Results and Discussion

### 3.1. Phase Observations

The two X-ray diffraction (XRD) patterns of the MnB nanoflakes are presented in Figure 2. It is evident that both MnB samples have an amorphous crystalline state since no distinctive diffraction peaks are present in their XRD patterns. In detail, the broad peak between 30° and 40° as well as between 50° and 70° in both of the XRD patterns can be assigned to the amorphous state of the Mn–B phase [23]. Their amorphous crystalline state was also confirmed by obtaining their corresponding SAED patterns as seen in Appendix A. A suggested reduction reaction mechanism for the formation of MnB nanoflakes is presented in reaction (1) as follows.
MnCl_2_ + 2NaBH_4_ + 3H_2_O → MnB +2NaCl + 5.5H_2_ + B(OH)_3_
(2)

Both sodium chloride (NaCl) and boric acid (B(OH)_3_) are soluble in water and are easily removed during the collection of the MnB precipitate through centrifugation followed by several rinses of the MnB powder with water.

### 3.2. Morphological Observations and Specific Surface Area Calculations

The TEM micrographs of the MnB nanoflakes are presented in Figure 3, and their high magnification micrographs are presented in the inserts of the same figure. The flaky morphology of the MnB nanoparticles is evident in both samples. When comparing the two samples with each other, both samples exhibit nanoflakes that are polydisperse in size, with the MnB−I sample having significantly smaller size flakes when compared to the MnB−II sample. This is also reflected in their calculated specific surface areas from their associated SAXS patterns (Figure 4). In addition, the thickness of the nanoflakes for sample MnB−I is very difficult to be determined from their TEM micrographs (Figure 3a). The same does not apply for the MnB−II sample where the nanoflakes are larger in size and have well-defined edges (Figure 3b). The average thickness of the MnB−II nanoflakes was calculated at ~5 nm. Lattice fringes are not present in any of the two samples, since the MnB nanoflakes are not crystalline enough. The calculated specific surface area for the MnB−I sample was 144 ± 14 m^2^/g (Figure 4a), whereas for the MnB−II sample it was 67 ± 7 m^2^/g (Figure 4b), almost half of the value of the MnB−I sample.

### 3.3. XPS Analysis

The chemical state of each element and the boron content in each MnB sample were investigated using X-Ray photoelectron spectroscopy (XPS) measurements. The atomic ratio of Mn:B for the MnB−I and MnB−II nanoflakes was calculated at 1.03:0.97 and 1.21:0.79, respectively, showing that the boron content decreases while the manganese content increases as more manganese ions are reduced with the increase in NaBH_4_ concentration during synthesis, as expected [16,24].

The Mn 2p spectra in both samples is composed of the Mn 2p 3/2 and Mn 2p 1/2 peaks (Figure 5a,c), which are related to spin orbit interactions or perhaps even coupling. In detail, three pairs of peaks were fitted for the Mn 2p spectra, suggesting the presence of three chemically distinct manganese states. The main pair of fitted peaks for sample MnB−I is located at binding energies of 641.2 eV and 653 eV and for sample MnB−II of 642 eV and 653.4 eV. This pair of peaks corresponds to the manganese bonded with boron (Mn-B) [25]. The next pair of fitted peaks for sample MnB−I is located at binding energies of 639.9 eV and 651.7 eV and for sample MnB−II of 643.9 and 652.7 eV. These peaks are related to the manganese-manganese bonds (Mn-Mn) [26]. The last pair of fitted peaks for sample MnB−I are at binding energies of 643.3 eV and 654.2 eV and for sample MnB−II of 643.5 eV and 654.7 eV. This pair of peaks are attributed to the manganese bonded to oxygen (Mn-O) [27]. On the other hand, the B 1s spectra for sample MnB−I and MnB−II are slightly different to each other as more chemical states of boron are present in MnB−II (Figure 5b,d). In detail, the B 1s spectra that corresponds to sample MnB−I, two peaks are present with the first peak located between 183.8 eV and 189.1 eV and the second located between 191 eV and 196 eV. The first peak was deconvoluted into two peaks, one with a maximum peak at 190.8 eV and the other at 191.8 eV. The first fitted peak corresponds to the elemental boron (B-B) and the second to the B-Mn species [25,28]. The second peak of the B 1s spectra with a maximum peak at 198.4 eV is assigned to the boron-oxo species [29,30]. For sample MnB−II three peaks are present in the B 1s spectra. The first beak is between 184.5 eV and 189.2 eV was deconvoluted into tow peaks with a maximum peak at 191.5 eV (B-B species) and 192.2 eV (B-Mn species). The second peak of the spectra has a maximum peak at 197.8 eV which corresponds to the B-O species, whereas the third peak of the spectra with a maximum peak at 199.3 eV is attributed to the B-O-Mn species [28].

### 3.4. Magnetic Properties

The zero dc-magnetic field and magnetic ac-susceptibility vs temperature smoothly increases as the temperature decreases (Figure 6a). For both samples, around 40 K and 10 K the real part of the magnetic ac-susceptibility, χ′(T), displays two small local maxima (see Figure 6b). The local maximum at around 10 K was attributed to the spin glass behavior [31]. Based on the results of Milivojevic et al. [32] the peak around 40 K can be attributed to λ-ΜnO_2_. In the ac-susceptibility measurements, under a 1 kOe dc-magnetic field, the local maximum of χ′(Τ) curve at T≈40 K disappeared, whereas the second one, T≈10 K, shifts slightly and becomes broader in comparison, and is observed for Hdc = 0. The imaginary part χ’’(T), is practically zero for both samples. A small peak is observed at T≈10 K (practically indistinguishable in the scale of Figure 6a,b).

The temperature dependence of the magnetic moment per g, σ(T), at a constant magnetic field Hdc = 1 kOe is depicted in Figure 6c. The σ vs T curves, for both samples show a step increase of σ around Τ≈44 Κ. A broader step increase, of σ(T) curves, is observed for both samples around T≈10 K. The ZFC and FC branches of the σ(T) (see Figure 6c) practically coincides.

The isothermal magnetic moment per g at T = 5 K, as a function of the external magnetic field, is illustrated in Figure 6d. For both samples, in the used dc-magnetic field, the magnetization does not saturate. Our σ(Hdc) data are similar with those reported in Ref. [31] for amorphous MnxB100-x alloys.

### 3.5. AFM Analysis

The topography of the MnB nanoflakes is shown in Figure 7. As shown in Table 2 and Figure 8, the resistance of the MnB−I sample was 1.80 times higher than that of the MnB−II sample, respectively, which means MnB−II possesses the highest electrical conductivity. The relative values presented here can be considered as reliable, whereas the absolute values depend on the contact area, which is not precisely known.

### 3.6. Electrocatalytic Properties

The electrocatalytic properties of the MnB nanoflakes were evaluated in a buffered electrolyte with 7 pH. A buffered electrolyte was used to make sure that the pH was stable throughout the electrochemical measurements for the proper HER and OER assessment [33].

#### 3.6.1. Hydrogen Evolution Reaction

The linear sweep voltammetry (LSV) for the HER to a current density of −10 mA cm^−2^ for the two MnB nanoflakes electrodes, as well as the LSV curve of the bare GC electrode are shown in Figure 9. According to the literature [34,35,36] −10 mA cm^−2^ is used for assessing the electrocatalytic activity, and, therefore, this value was chosen.

In neutral conditions (pH 7) both the H_2_O molecules and the hydronium cation H_3_O^+^ ions participate in the HER mechanism in a two-step reduction process [37]:

(1)At low cathodic overpotentials close to the equilibrium potential (i.e., 0 V vs. reversible hydrogen electrode (RHE)), H_3_O^+^ ions are the species that dominate;
H* + H_3_O^+^ + e^−^ → H_2_ + *(3)(2)At high cathodic overpotentials, H_3_O^+^ ions convert to H_2_O molecules.
H* + H_2_O + e^−^ → H_2_ + OH^−^(4)

In the above reactions, * represents the catalyst surface, and H* represents the adsorbed hydrogen intermediate. The MnB−I exhibited the lowest potential at a current density of −10 mA cm^−2^ (Figure 9a). The same sample also had the smallest empirical Tafel slope (Figure 9b) when compared to the MnB−II. This shows that the HER mechanism in neutral conditions is dictated by the specific surface area of the catalyst and not as much by the electric conductivity, since this sample was not as conductive as the MnB−II sample. A high specific surface area results in more active sites on the catalyst’s surface and therefore better electrocatalytic activity. The Tafel slope is linked to the intrinsic nature of the electrocatalyst. The probable HER mechanisms, and the possible kinetic rate of these reactions can be determined by the empirical values of the slope [37,38]. The empirical values of the Tafel slopes for the two MnB nanoflake samples (Figure 9b) show that the Volmer reaction takes place. The MnB−I sample exhibited the fastest reaction kinetics in comparison to the MnB−II sample. Both samples generated hydrogen with the same HER mechanisms regardless of their physicochemical differences. The MnB−I sample had the highest specific surface area, the highest HER reaction kinetics, and the lowest overpotential.

#### 3.6.2. Oxygen Evolution Reaction

Figure 10 displays LSV for the OER up to a current density of 10 mA∙cm^−2^ for the two MnB nanoflake samples and the bare GC electrode.

In neutral conditions (pH 7), the OER mechanism takes place as seen below:2H_2_O + * → OH* + H_2_O + e^−^ + H^+^(5)
OH* + H_2_O → O* + H_2_O + e^−^ + H^+^
(6)
O* + H_2_O → *OOH + e^−^ + H^+^(7)
*OOH → O_2_ + e^−^ + H^+^(8)

In Figure 10, the MnB−II sample showed the lowest potential at a current density of 10 mA cm^−2^, and the smallest empirical values of the Tafel slope when compared to the MnB−I sample. Considering the physiochemical properties of these samples, it is evident that the electronegative manganese atoms are the active sites on the catalyst surface that contribute to achieving an enhanced catalytic performance compared to the specific surface area. This heightened catalytic performance is linked as follows: (i) reduction in the reaction’s thermodynamic barrier (reaction (7)) for the production of an oxygen molecule from the coupling of the two adsorbed oxygen atoms on the catalysts surface, (ii) reduction in the reaction’s thermodynamic barrier for the formation of the *OOH intermediate, and (iii) the enhancement of the reaction kinetics.

#### 3.6.3. Stability

One hundred scans at a scan rate of 1 mVs^−1^ were performed to evaluate the stability of the MnB nanoflakes for HER and OER at pH 7. Figure 11 demonstrates how the potential at a current density of 10 mA cm^−2^ fluctuates over the consecutive scans. Both the MnB−I and MnB−II samples (Figure 11a) exhibited a stable HER activity over 100 scans, demonstrating no degradation. The OER stability test for the MnB−II sample (Figure 11b) showed quite stable behavior throughout. The MnB−I sample exhibited a drop in its OER activity over time as its conductive layer peeled off locally from the electrode’s surface after the first scan. The good HER and OER stability of the MnB−II sample for the HER and the OER could be due to its enhanced conductivity and/or the reversible electron transfer phenomena in the amorphous manganese boride samples. The electronegative Mn active sites present on the MnB−II sample may avert the Mn sites from oxidation, enhancing in this way the overall catalyst stability.

## 4. Conclusions

The 2D manganese mono-boride nanoparticles with a flaky amorphous structure were successfully synthesized at room temperature by a simple one-pot chemical reduction method. The sample with the lowest-boron content exhibited the highest specific surface area, better magnetic properties, and electrocatalytic activity in the HER. Whereas the sample with the highest-boron content was more conductive and demonstrated better electrocatalytic properties in the OER. This demonstrates that material properties, such as particle size and the chemical state need to be considered in detail when designing nanomaterials for specific application.

## Figures and Tables

**Figure 1 nanomaterials-13-00300-f001:**
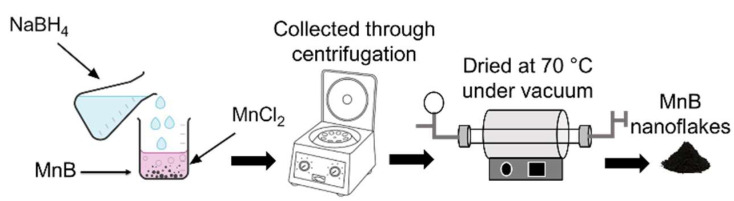
Synthesis procedure for the MnB nanoflakes.

**Figure 2 nanomaterials-13-00300-f002:**
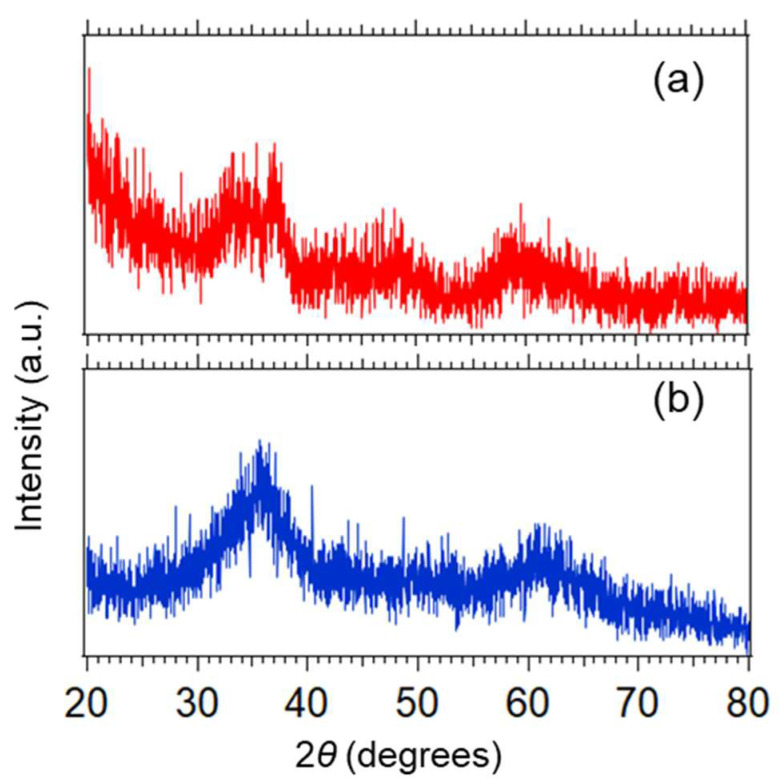
XRD patterns of (**a**) MnB−I and (**b**) MnB−II.

**Figure 3 nanomaterials-13-00300-f003:**
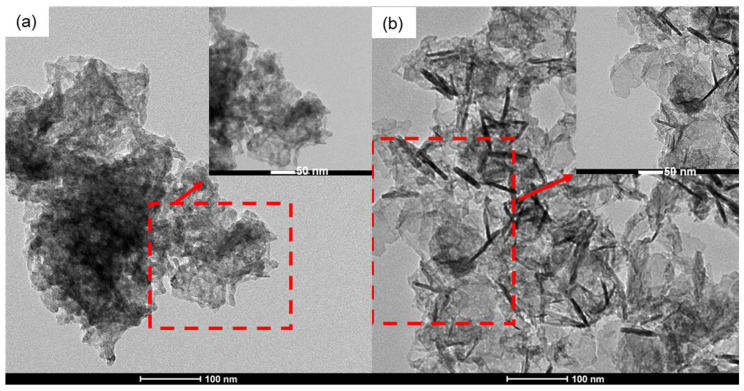
TEM micrographs of the (**a**) MnB−I and (**b**) MnB−II nanoflakes.

**Figure 4 nanomaterials-13-00300-f004:**
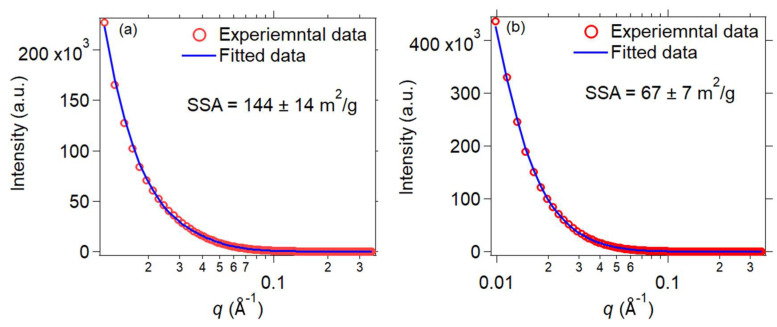
SAXS patterns of the MnB−I (**a**) and MnB−II (**b**) nanoflakes. The red circles are the experimental data, and the solid blue line is the fitted data using a unified fit in Irena. *B* was 0.07 and 0.03 cm^−1^ Å^−4^ for MnB−I and MnB−II, respectively. *δ* was 2.57 g/cm^3^ and Δ*ρ*^2^ was 396∙10^20^ cm^−4^.

**Figure 5 nanomaterials-13-00300-f005:**
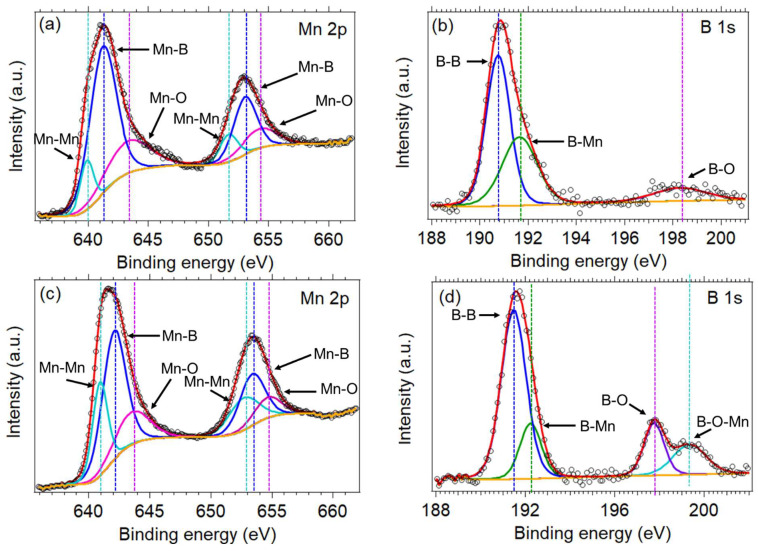
Mn 2p XPS and B 1s spectra of the (**a**,**b**) MnB−I and (**c**,**d**) MnB−II. The black circle markers represent the experimental intensity, the red line represents the fitted data, and the yellow solid line represents the background.

**Figure 6 nanomaterials-13-00300-f006:**
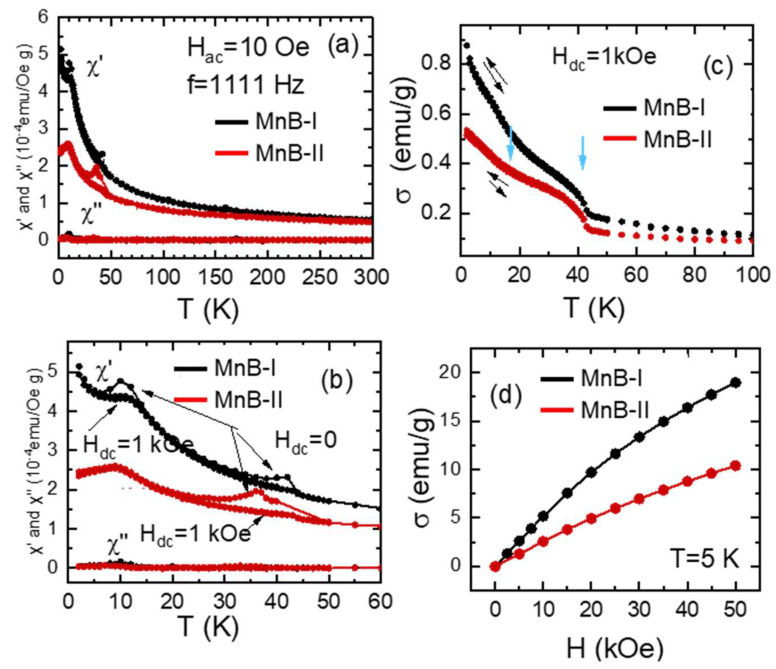
(**a**) Real χ′(T) and imaginary χ″(T) magnetic ac-susceptibility, as a function of temperature, of the samples MnB−I and MnB−II.; (**b**) details of the magnetic ac-susceptibility measurements in the temperature interval T = 0–60 K; (**c**) temperature variation in the dc-magnetic moment per g, of MnB−I and MnB−II samples, measured under 1kOe external magnetic field following the ZFC and FC protocols; (**d**) isothermal at T = 5 K variation in the magnetic moment/g for MnB−I and MnB−II samples.

**Figure 7 nanomaterials-13-00300-f007:**
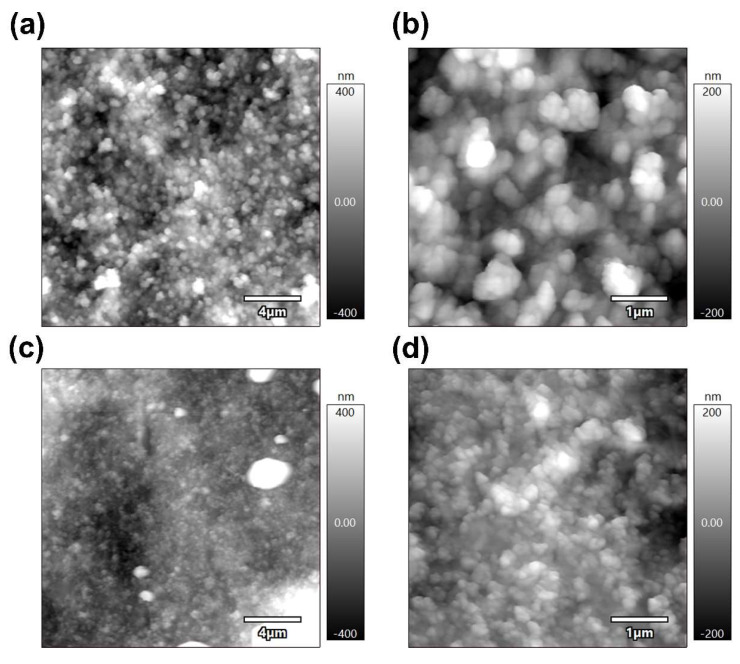
Amplitude modulation mode topography images of (**a**,**b**) MnB−I and (**c**,**d**) MnB−II.

**Figure 8 nanomaterials-13-00300-f008:**
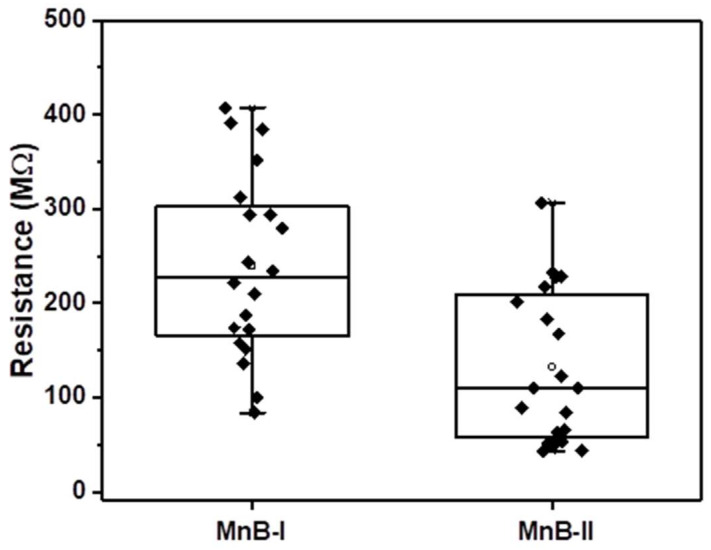
Box plot of resistance of each sample/region as determined from C-AFM IV measurements (*n* = 20).

**Figure 9 nanomaterials-13-00300-f009:**
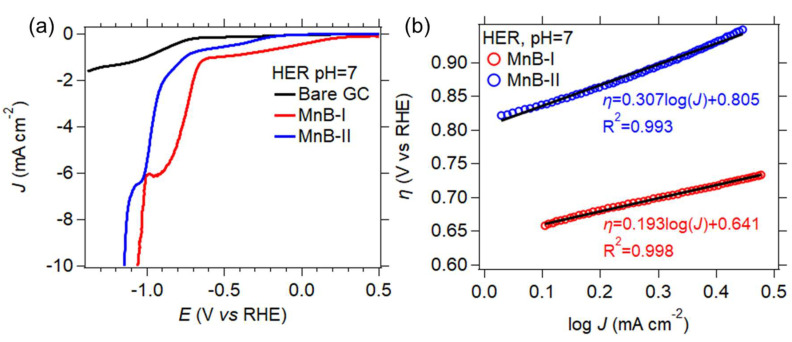
(**a**) Current density (*J*) vs. potential (*E*) linear polarization curves obtained from hydrogen evolution linear sweep voltammetry (LSV) scan at 1 mVs^−1^ in pH 7 electrolyte; (**b**) HER Tafel plots of all samples in electrolytes with a pH 7 derived from the linear polarization curves. Linear regressions are represented by the black lines, and the equations of the linear fits are also shown.

**Figure 10 nanomaterials-13-00300-f010:**
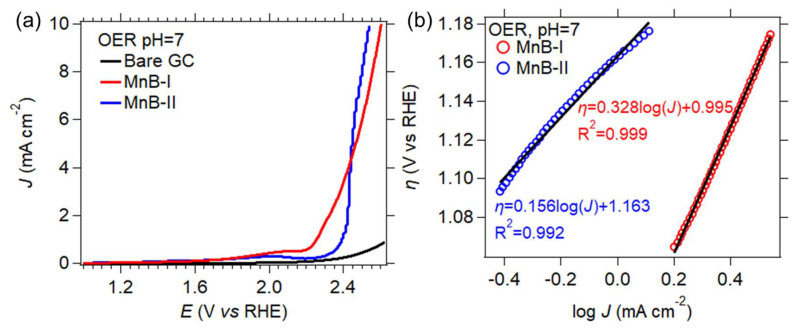
(**a**) Current density (*J*) vs. potential (*E*) linear polarization curves obtained from hydrogen evolution linear sweep voltammetry (LSV) scan at 1 mV∙s^−1^ in pH 7 electrolyte; (**b**) HER Tafel plots of all samples in electrolytes with a pH 7 derived from the linear polarization curves. Linear regressions are represented by the black lines, and the equations of the linear fits are also shown.

**Figure 11 nanomaterials-13-00300-f011:**
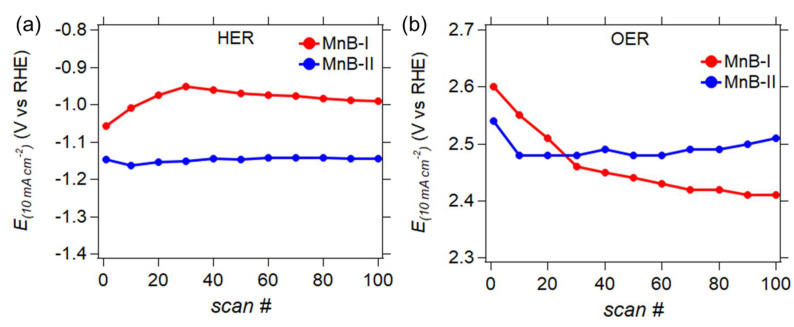
Potentials at +10 A∙m^−2^ (*E*_(*J* = −10 A m^−2^)_) vs. consecutive LSV scan number (*scan #*), demonstrating the stability of the response over 100 scans for (**a**) HER and (**b**) OER.

**Table 1 nanomaterials-13-00300-t001:** Synthesis description and sample names used in this study.

Description	Sample Name
NaBH_4_ (0.5 M) + MnCl_2_ (0.27 M)	MnB−I
NaBH_4_ (1 M) + MnCl_2_ (0.27 M)	MnB−II

**Table 2 nanomaterials-13-00300-t002:** The resistance of each sample/region as determined from C-AFM IV measurements (*n* = 20).

Sample	Resistance (MΩ)
MnB−I	239.3 ± 91.7
MnB−II	132.6 ± 81.0

## Data Availability

The data presented in this study are available on request from the corresponding author.

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
