# Peer review of "Exploring the Magnetic and Electrocatalytic Properties of Amorphous MnB Nanoflakes"

_nanomaterials, 2023, doi:10.3390/nano13020300_

Round 1
Reviewer 1 Report
The authors reported the synthesis of manganese mono-boride nanoflakes. The material characterization is in detail. The material is of interest and versatile for magnetic and electrocatalytic properties. I think this paper can be considered for publication after the authors addressed following issues.
Specific comments:
1. The author claimed that the MnB samples exhibit the amorphous crystalline from the XRD results, please give the SAED pattern.
2. The HER or OER performance should be compared with the previously reported papers.
3. Some papers may be cited, such as Nano Research Energy 2022, 1: e9120010; Journal of Alloys and Compounds, 2022, 929, 167254.
4. The Rs or Rct values of MnB are two large. Please confirm that.
Reviewer 2 Report
A single manganese boride nanocrystal was synthesized by one - pot chemical reduction method at room temperature. The effects of specific surface area and boron content on magnetic and electrocatalytic properties of single manganese boride nanocrystals were reported. The importance of basic and practical parameters that must be considered when designing nanomaterials for specific applications is emphasized. This work can be accepted for publication after major corrections. My comments are given below:
(1) The abstract mentions that "The sample with the highest specific surface area and boron content demonstrated the best magnetic and magnetic electrocatalytic properties in the HER. Whereas the sample with lowest specific surface area and boron content exhibited the highest electric conductivity and electrocatalytic properties in the OER. "If only two samples" best "use is appropriate.
(2) the "lowest" mentioned in the previous question should be revised to "the lowest".
(3) The article mentioned that "nanoflakes influence their magnetic and electrocatalytic properties". Only two ratios can be well verified
(4) Whether the proportion of precursor of the preparation method is directly related to the proportion of elements in the product.
(5) As for the analysis of XRD mentioned in Figure 2, it is believed that "the broad peak between 30° and 40° in both XRD patterns can be assigned to the amorphous state of the Mn-B phase." What crystalline material is the 50° to 70° peak.
(6) Is it reasonable to express stability in the way shown in Figure 11? Which test method was referred to? Whether lsv comparison before and after the reaction is required.
(7) What are the advantages of sample performance compared with other literature?
(8) Some important and the latest related research progresses, such as Nano Energy 104 (2022) 107875 for electrochemical HER and OER can be introduced to let the authors understand the background especially the details of electrochemistry related tests well.

Round 2
Reviewer 2 Report
The manuscript has been sufficiently improved to warrant publication in Nanomaterials.